# Structural evolution of nitrogenase states under alkaline turnover

Rebeccah A. Warmack [1] ✉ & Douglas C. Rees [1,2] ✉

Biological nitrogen fixation, performed by the enzyme nitrogenase, supplies nearly 50% of the bioavailable nitrogen pool on Earth, yet the structural nature of the enzyme intermediates involved in this cycle remains ambiguous. Here we present four high resolution cryoEM structures of the nitrogenase MoFe-protein, sampled along a time course of alkaline reaction mixtures under an acetylene atmosphere. This series of structures reveals a sequence of salient changes including perturbations to the inorganic framework of the FeMo-cofactor; depletion of the homocitrate moiety; diminished density around the S2B belt sulfur of the FeMo-cofactor; rearrangements of cluster-adjacent side chains; and the asymmetric displacement of the FeMo-cofactor. We further demonstrate that the nitrogenase associated factor T protein can recognize and bind an alkaline inactivated MoFe-protein in vitro. These time-resolved structures provide experimental support for the displacement of S2B and distortions of the FeMo-cofactor at the $E_0$-$E_3$ intermediates of the substrate reduction mechanism, prior to nitrogen binding, highlighting cluster rearrangements potentially relevant to nitrogen fixation by biological and synthetic clusters.

Atmospheric dinitrogen contains one of the most stable bonds found in nature[1] and undergoes only one biological conversion. This reaction produces metabolically available ammonia through the reduction mechanism of the enzyme complex nitrogenase found in a division of prokaryotes known as diazotrophs. In the last century, the industrial Haber–Bosch process has matched these natural levels of ammonia production through a process requiring high temperatures and pressures, making a significant overall contribution to $CO_2$ emissions[2]. In contrast, the nitrogenase system accomplishes the conversion at ambient temperatures and pressures through the action of two component proteins: the catalytic MoFe-protein and the obligate reductase Fe-protein[3–5]. Electrons for substrate reduction by nitrogenase are derived in vivo from flavo- or ferredoxins and are transferred in an ATP-dependent fashion through the [4Fe:4S] cluster of the Fe-protein to the MoFe-protein[6,7]. The MoFe-protein is a heterotetramer with the homologous α- and β-subunits forming an $α_2β_2$ dimer of αβ dimers. Sandwiched between the two subunits within an αβ dimer lies the intermediary [8Fe:7S] P-cluster. This cluster transfers electrons into

the active site [7Fe:1Mo:9S:C]-*R*-homocitrate FeMo-cofactor positioned within the α-subunit. One electron is transferred from the Fe-protein to the MoFe-protein per two ATP hydrolyzed, and multiple cycles result in increasingly reduced '$E_n$' states of the MoFe-protein that are able to bind and reduce substrates[8,9] (Supplementary Fig. 1).

Despite the presence of eight metal sites in the FeMo-cofactor, the cluster is coordinated to the protein through the side chains of only two residues, α-Cys275 and α-His442, that serve as ligands to the terminal Fe1 and Mo sites, respectively. The coordination of Mo is completed by the *R*-homocitrate (HCA) ligand bound in a bidentate manner as identified in resting state crystal structures[10]. While the sparse coordination of the FeMo-cofactor may be postulated to be advantageous for cluster insertion[11], in the similarly sized P-cluster, each metal has a protein ligand. The distinct coordination of the FeMo-cofactor is accompanied by a notable lability of its three belt sulfurs, as demonstrated by their displacement by carbon monoxide (CO) and selenium under certain reaction conditions, as well as a carbonate ligand in the cofactor of the vanadium (VFe) nitrogenase[12–14]. While CO

[1]Division of Chemistry and Chemical Engineering 147-75 California Institute of Technology, Pasadena, CA, USA. [2]Howard Hughes Medical Institute, California Institute of Technology, Pasadena, CA, USA. ✉e-mail: rwarmack@caltech.edu; dcrees@caltech.edu

specifically displaces belt sulfur S2B, turnover with selenocyanate has been found to replace all three belt sulfurs (S2B, S3A, and S5A) to varying extents with selenium. Of note, crystallographic studies of the VFe nitrogenase identified a light atom ligand replacing sulfur S2B, with a sulfur-containing density identified within a pocket created by the movement of the active site α-Gln176 residue (equivalent to α-Gln191 in MoFe nitrogenase)[15].

In addition to changes at the belt sulfur sites in the FeMo-cofactor, perturbations to the coordination of the Mo-HCA have recently been found in asymmetrically disordered cryoEM structures prepared under turnover conditions at neutral pH and in an anaerobic turnover-inactivated state at high pH[16,17]. HCA-deficient, disordered states of the MoFe-protein have also been further observed bound to an endogenous protein partner when purified from *Azotobacter vinelandii* cells, the nitrogenase associated factor T (NafT)[17], suggesting that these altered states are physiologically relevant. These findings highlight the important role of HCA, a potentially labile ligand, in the nitrogenase mechanism[18]; indeed, changes at the HCA-Mo coordination site during turnover have been previously proposed to generate a binding site for substrates[19–21]. The observed lability of HCA from the active site under turnover at high pH[17] raises the question of possible changes in cofactor coordination during catalysis and the implications these may have on the overall architecture of the active site. Access to structures of catalytic intermediates obtained under turnover conditions would expand our understanding of how such transformations take place.

Nitrogenase is a relatively slow enzyme (with turnover times derived from specific activity measurements ranging from a few hundred milliseconds for proton and acetylene reduction to about a second for dinitrogen, based on the comparable specific activities for the *Klebsiella pneumoniae* and *A. vinelandii* enzymes[22]. Nonetheless, the capture of nitrogenase structural dynamics is a challenge for crystallographic approaches given the multiple intermediates and the kinetics of their formation and interconversion[9]. Further, nitrogenase cannot be studied in the absence of reducible substrates since protons are always present and the reduction of most substrates is in competition with proton reduction to dihydrogen. As an approach to overcoming some of these problems, cryoEM offers an opportunity to sample time-dependent structural differences during turnover by enabling sample freezing in as little as 54 ms using commercially available equipment[23]. This aids in the capture of sufficiently long-lived, turnover-associated states. Further, altered biochemical conditions can facilitate the probing of long-lived intermediates by slowing the nitrogenase reaction; such is the case of nitrogenase reactions at high pH which lowers the proton concentration required for substrate reduction[24,25].

In this study, we investigate the structural states of nitrogenase that exist at different times under alkaline (pH 9.5) turnover conditions in the presence of acetylene. Acetylene reduction offers two powerful advantages for these studies relative to dinitrogen reduction: (i) acetylene reduction to ethylene is a two electron process rather than eight electrons, as for dinitrogen reduction with obligatory hydrogen evolution, so that there are fewer intermediate states[26] and (ii) acetylene reduction fully suppresses hydrogen evolution so that the electrons are funneled exclusively into ethylene[27] (Supplementary Fig. 1). Under alkaline (pH 9.5) conditions, the rate of substrate reduction decreases[25] potentially extending the time for the reaction to reach steady state. Further, as we previously demonstrated, the MoFe-protein becomes inactivated under turnover conditions at alkaline pH (pH 9.5), as indicated by a decrease in the rate of ethylene produced[24]. Although the loss of catalytic activity at pH 9.5 follows first order kinetics with a half-life corresponding to ~10 min, the kinetics of irreversible inactivation are complex and pH-dependent; complete inactivation of the MoFe-protein as monitored by a specific activity assay at pH 7.8 requires 4–6 h[24].

To illuminate structures along this pathway, here we determine four distinct cryoEM structures of the nitrogenase MoFe-protein, sampled anaerobically along a time course, directly from alkaline (pH 9.5) acetylene nitrogenase turnover mixtures. In this sequence, an increasing amount of disorder is observed within the MoFe-protein αIII domain, primarily in an asymmetric fashion as observed under turnover in the presence of dinitrogen[16]. With increasing time, structures show distortions within the MoFe-protein active site including perturbations to the inorganic framework of the FeMo-cofactor, depletion of the HCA and S2B belt sulfur densities, and the asymmetric displacement of the FeMo-cofactor in the active site. These results shed light on how changes at the FeMo-cofactor may take place during the early steps of nitrogenase substrate turnover.

## Results
### Time series structures of the MoFe-protein during acetylene turnover
We monitored the turnover process by plunge freezing aliquots of the alkaline (pH 9.5) reaction mixture directly from vials onto cryoEM grids, within an anaerobic chamber (Supplementary Fig. 1)[28]. Time points were chosen above and below the previously estimated rate of inactivation with a $t_{1/2}$ of ~10 min[24], and before the complete inactivation observed at 4 h. Thus, reactions were sampled at 20 sec, 5 min, 20 min, and 60 min time points, yielding four structures (Fig. 1A, B and Supplementary Fig. 2). These structures are termed the MoFe$^{Alkaline-20sec}$, MoFe$^{Alkaline-5min}$, MoFe$^{Alkaline-20min}$, and MoFe$^{Alkaline-60min}$ states. Specific activity measurements of ethylene production at pH 7.8 demonstrate that the MoFe$^{Alkaline-20sec}$, MoFe$^{Alkaline-5min}$, MoFe$^{Alkaline-20min}$, and MoFe$^{Alkaline-60min}$ states retain 91%, 62%, 44%, and 17% of activity, respectively (Fig. 1C). The production at 20 sec of approximately 1 nmol ethylene by 0.25 mg of MoFe-protein (~1 nmol protein or 2 nmol active site), indicates that the earliest time point captures the initial turnover event. The structures were resolved to nominal resolutions between approximately 1.9 and 2.2 Å and have allowed us to sample the inactivation process with sufficient resolution to identify molecular-resolution changes in the MoFe-protein (Fig. 1D; Supplementary Figs. 2 and 3; Supplementary Table 1).

A previously determined cryoEM structure of MoFe-protein isolated from alkaline (pH 9.5) acetylene reduction reactions in the absence of ATP, termed MoFe$^{Alkaline}$ ($t = 0$), revealed a nearly identical structure to the crystallographic resting state at physiological pH (pH 7.8)[17]. In contrast, the first three time resolved structures reveal a progressive, asymmetric disordering of density within the two MoFe-protein α-subunits as a function of time under acetylene turnover (Fig. 1A, B and Supplementary Fig. 4). These changes are primarily isolated to the αIII domain within that subunit as noted in previous studies[16,17], with residues 1–50, 355–361, and 380–409 being invariably disordered within at least one αβ dimer throughout all the sampled reaction time points beyond $t = 0$ (Fig. 1A, B). Asymmetric disordering within the αIII domain was previously observed in cryoEM structures under N$_2$ turnover[16]. In contrast, at the last time point, MoFe$^{Alkaline-60min}$, the observed disordering within the α-subunit is nearly symmetric between the two αβ dimers. Collectively, these changes suggest that αIII domain disordering precedes catalytic inactivation (Fig. 1).

### Changes in the ordered active site cofactor during turnover
The MoFe$^{Alkaline-20sec}$, MoFe$^{Alkaline-5min}$, and MoFe$^{Alkaline-20min}$ structures all contain one well-ordered active site in which the FeMo-cofactor and HCA occupy resting state positions (Fig. 2A–D). Interestingly, when the cryoEM density within the active sites of these more ordered αβ dimers is compared, a depletion in density can be seen specifically around the FeMo-cofactor belt sulfur S2B site relative to the shortest time point (Fig. 2E–G). This observation is corroborated by volume measurements of the cryoEM density surrounding the S2B atom, while the volumes of the cryoEM density about the other belt sulfurs remains

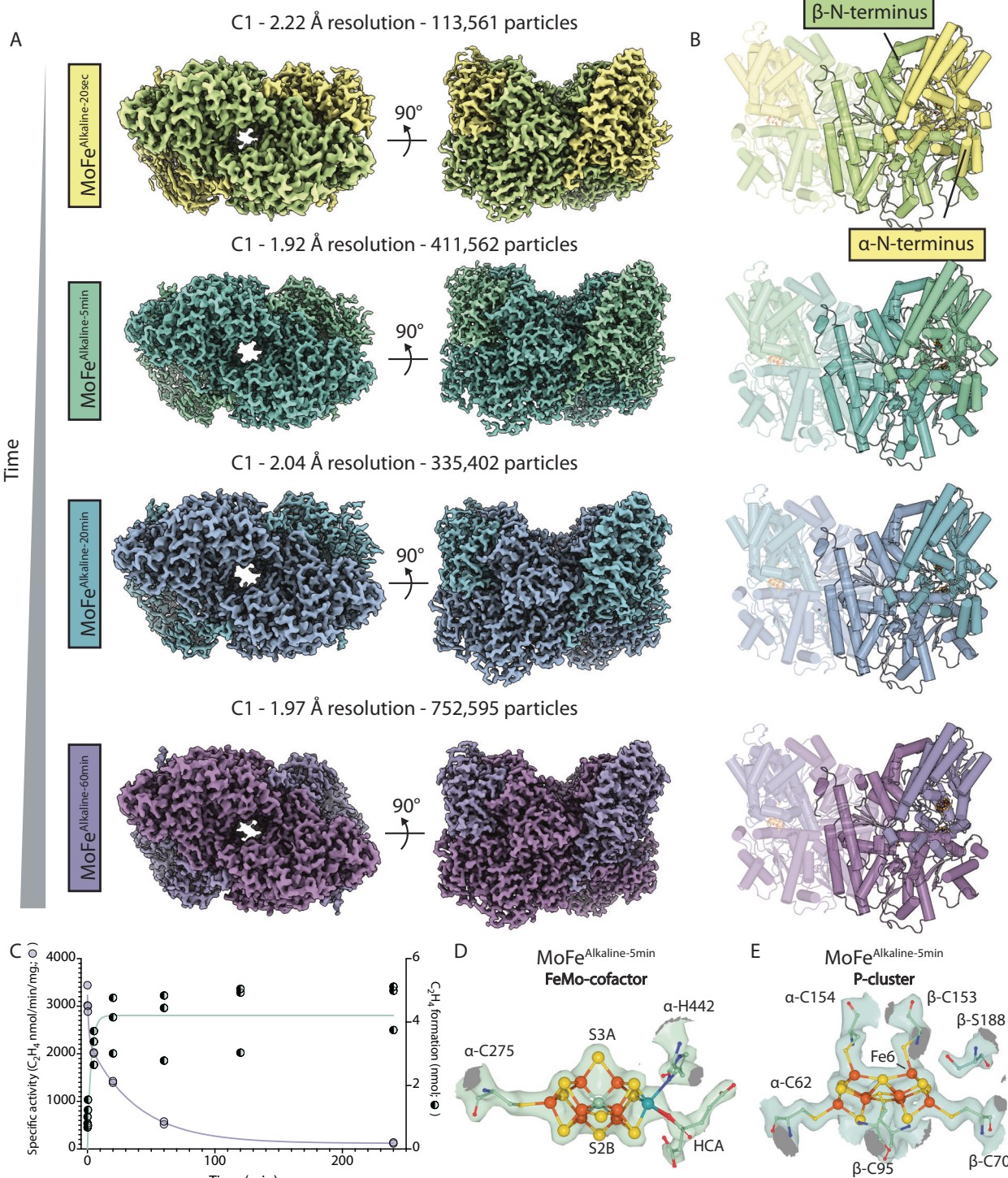

**Fig. 1 | Time sequence cryoEM structures of the nitrogenase MoFe-protein under alkaline turnover.** **A** CryoEM density maps of MoFe-protein from acetylene reaction mixtures at pH 9.5 at 20 s (yellow; MoFe$^{Alkaline-20sec}$), 5 min (green; MoFe$^{Alkaline-5min}$), 20 min (blue; MoFe$^{Alkaline-20min}$), and 60 min (purple; MoFe$^{Alkaline-60min}$). The coloring convention of each time point will be maintained throughout the manuscript. **B** Structural models of the adjacent cryoEM maps. α- and β-subunits are highlighted in the figure panel. **C** Left axis: Specific activity assays of acetylene turnover to ethylene under alkaline conditions (solid symbols; purple line is fit according to a two phase exponential decay model for illustrative purposes only;

$n = 2$ technical replicates). Right axis: Total ethylene formation assays under alkaline (pH 9.5) conditions (striped symbols; green curve is fit according to equation 1 in Yang et al.[24]; $n = 3$ technical replicates). Assay conditions contained ~1 nmol of MoFe-protein, or ~2 nmol of active site. Source data are provided as a Source Data file. **D** CryoEM density for the ordered active site FeMo-cofactor in the MoFe$^{Alkaline-5min}$ time point. **E** CryoEM density for the P-cluster in the MoFe$^{Alkaline-5min}$ time point. For D and E, atoms are colored according to the following scheme: green (carbon); red (oxygen); blue (nitrogen); yellow (sulfur); orange (iron); teal (molybdenum).

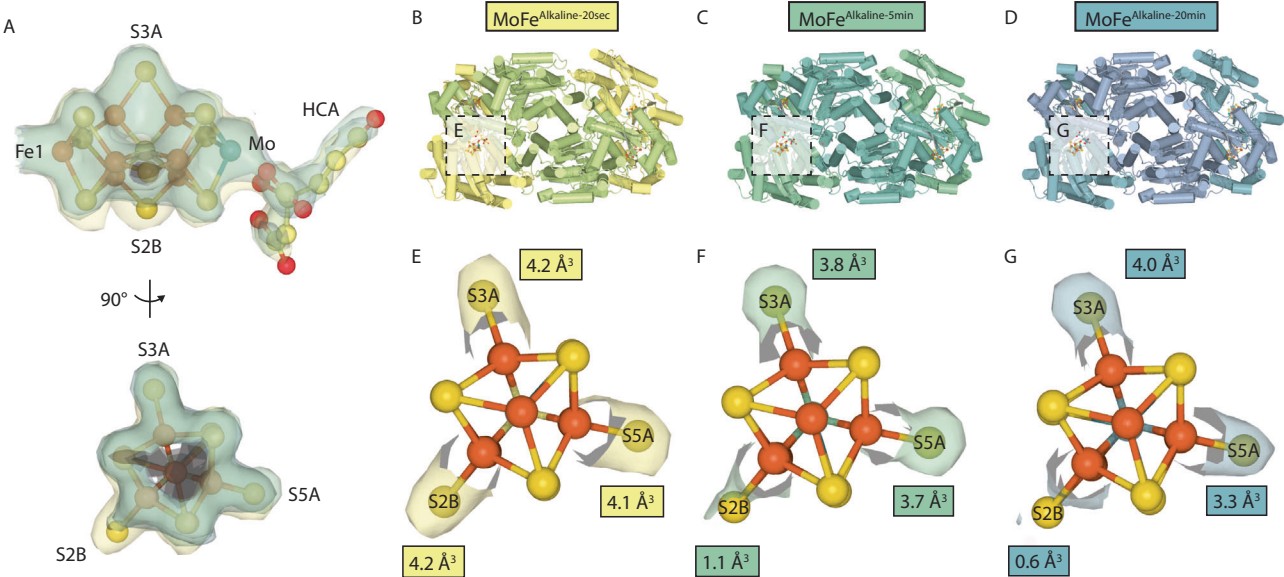

**Fig. 2 | Time-dependent sulfur S2B depletion in the ordered MoFe-protein active sites. A** Overlay of cryoEM density (12 σ) surrounding the ordered FeMo-cofactor and homocitrate (HCA) for the MoFe[Alkaline-20sec] (yellow), MoFe[Alkaline-5min] (green), and MoFe[Alkaline-20min] (blue) time points. The belt sulfurs of FeMo-cofactor S3A, S5A, and S2B are labeled. **B–D** Structures of the MoFe[Alkaline-20sec], MoFe[Alkaline-5min], and MoFe[Alkaline-20min] times points, which each contain one ordered active site (boxed). **E–G** Volumes of cryoEM density around the FeMo-cofactor belt sulfur atoms were calculated at 15 σ within a 1.25 Å spherical radius; values are shown in boxes adjacent to the corresponding belt sulfur.

relatively constant, though a slight decrease is noted in the S5A density (Fig. 2G)[13]. The VFe nitrogenase structure after turnover revealed the presence of a light (O/N) atom replacing sulfur at the S2B position[15], with the displaced sulfur likely bound in a pocket created by the movement of α-Gln176, which in the Mo-nitrogenase is α-Gln191. Despite the observed decrease in S2B density in the ordered active sites of MoFe[Alkaline-5min] and MoFe[Alkaline-20min] relative to MoFe[Alkaline-20sec], there is no corresponding movement of the α-Gln191 side chain in these more ordered active sites, nor is there extraneous density observed that might indicate the presence of a displaced sulfur (Supplementary Fig. 5).

**Changes in the disordered active site cofactor during turnover**
Progressive decreases in the elution time of the MoFe-protein by S.E.C. were observed when reaction mixtures were inhibited with excess MgADP at the 20 sec, 5 min, 20 min, and 60 min time points, possibly reflecting time-dependent increases in the hydrodynamic radius (Fig. 3A). This is consistent with the disordering observed within the α-subunit of at least one αβ dimer in the EM structures of all four time points (Fig. 3B–E). The earliest timepoint structure, MoFe[Alkaline-20sec], already reveals a disordered αIII domain with the associated FeMo-cofactor lacking recognizable density for HCA (Fig. 3F). These perturbations notwithstanding, cryoEM density clearly outlines the location of the FeMo-cofactor in this active site in the same location as the resting state. Intriguingly, when modeled with the resting state FeMo-cofactor, the density reveals changes near the S5A-Fe7 bond (Fig. 3F). Density around Fe7 suggests an outward displacement of this atom away from the central carbide, elongating the typical 2.0 Å Fe-C bond distance to over 2.5 Å. While modeling density around metalloclusters at lower resolutions can be challenging, we note that computational studies have suggested perturbations to the inorganic framework of the FeMo-cofactor accompanying protonation of the interstitial carbide[29,30]. A model of this state proposed by McKee[29] better fits the altered FeMo-cofactor density of the MoFe[Alkaline-20sec] state than the resting state FeMo-cofactor (Supplementary Fig. 6). Unassigned density is additionally observed stemming from S2B between α-His195 and α-Gln191. While an acetylene molecule can be placed into this density,

the resolution is insufficient to definitively identify the atoms in this density, thus it has been left unmodelled in the deposited structure.

The disordered active sites across all sampled time points lack intact HCA density (Fig. 3F–J). Time points later than 20 sec also show substantially diminished and fractured density for the FeMo-cofactor, reminiscent of the broken density observed within the previously determined MoFe[Alkaline-inactivated] state active site[17] (Fig. 3G–J). Interestingly, the density corresponding to these cofactors also seems to be consistently broken near the Fe7 site, with a weakening of the S5A density, overlapping with the distortions seen in the MoFe[Alkaline-20sec] state (Fig. 3F), albeit the current local resolution limits further interpretation. By the 60 min time point, the two αIII domains of MoFe[Alkaline-60min] are nearly symmetrically disordered and both active sites of MoFe[Alkaline-60min] lack HCA density. However, the FeMo-cofactors appear better ordered in this final time point, likely representing a more homogeneous mixture of particles. The density for the disordered active sites of MoFe[Alkaline-60min] support a rearrangement of the α-His442 loop as seen in the MoFe[Alkaline-inactivated] structure and the ΔnifB MoFe-protein crystal structure (Fig. 3I, J and Supplementary Fig. 5)[11,17].

In these disordered active sites, density around the belt sulfur atoms limits interpretation, but the α-Gln191 residue clearly rearranges in a time-dependent manner. Associated with this rearrangement is the appearance of a new, spherical density (Fig. 4). The repositioning of α-Gln191, and the appearance of a new density is strongly reminiscent of the VFe nitrogenase turnover crystal structure which revealed a light atom in the place of S2B, the change in α-Gln176 (α-Gln191 in MoFe nitrogenase) conformation, and the formation of a sulfur binding pocket verified by anomalous scattering[15]. While it is tempting to assign the new density as the (hydro)sulfide binding site, we cannot confirm the identity of this spherical density at the resolution of these maps and have left this density modeled as a water molecule in the deposited structures.

**Changes outside the active site**
The P-cluster of the MoFe-protein can adopt three conformations depending on the oxidation state: $P^N$, $P^{1+}$, and $P^{2+}$, with $P^N$ representing the fully reduced state[31,32]. Upon oxidation, the $P^{1+}$ and $P^{2+}$ states adopt

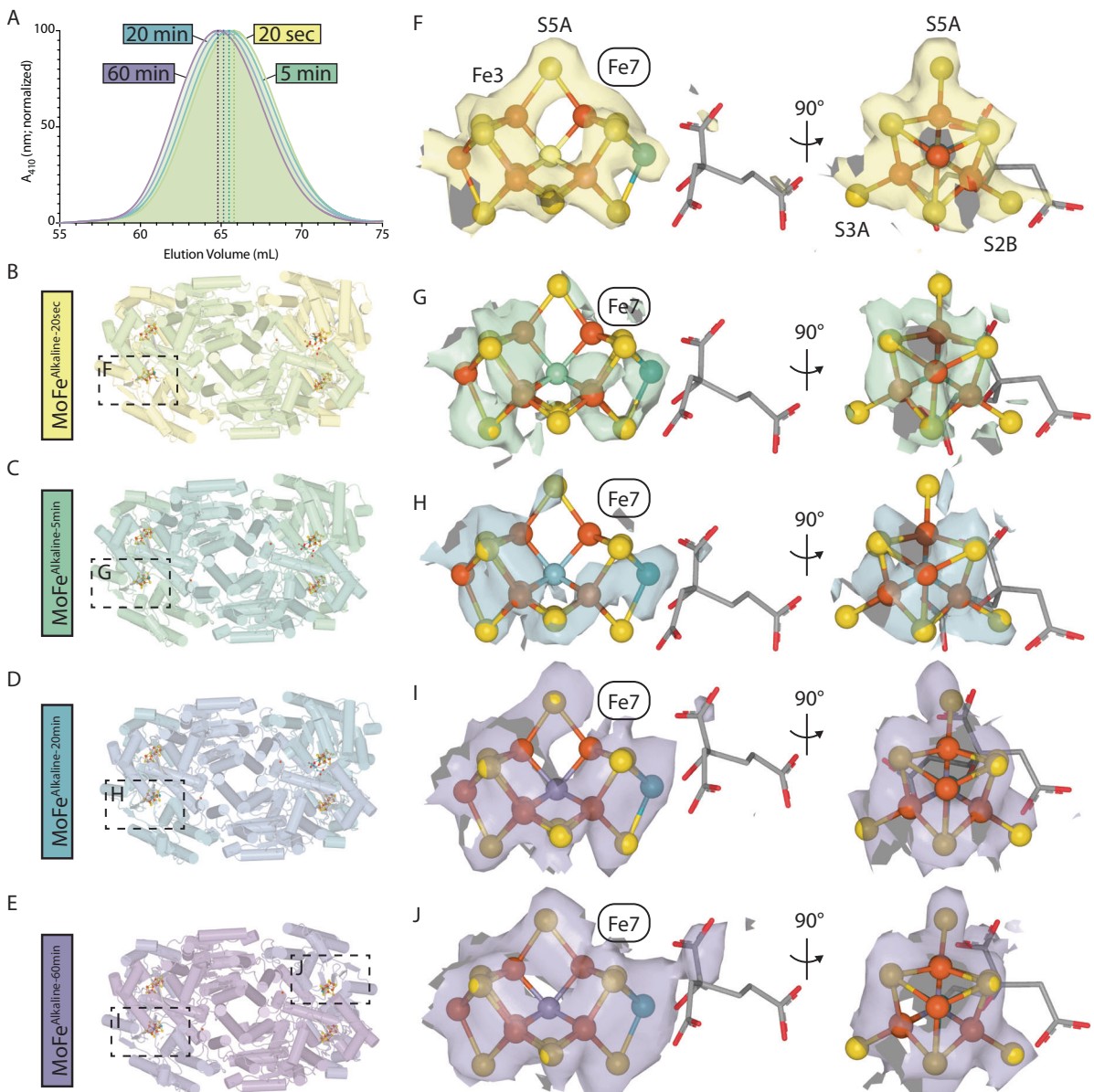

**Fig. 3 | Time-dependent FeMo-cofactor rearrangement in the disordered MoFe-protein active sites. A** Size exclusion chromatography (S.E.C.) of reaction mixtures stopped at the designated time points with addition of excess MgADP. Dotted lines indicate the peak elution time for the MoFe-protein from each reaction mixture: 65.8 min (20 sec; yellow), 65.6 min (5 min; green), 65.2 min (20 min; blue), 64.9 min (60 min; purple). Source data are provided as a Source Data file. **B–E** Structure models of the MoFe$^{Alkaline-20sec}$ (yellow), MoFe$^{Alkaline-5min}$ (green), MoFe$^{Alkaline-20min}$ (blue), and MoFe$^{Alkaline-60min}$ (purple) time points with disordered active sites indicated with boxes. These panels are rotated 180° relative to the orientation depicted in Fig. 2B–D. **F–J** CryoEM density surrounding the disordered FeMo-cofactor for the MoFe$^{Alkaline-20sec}$, MoFe$^{Alkaline-5min}$, MoFe$^{Alkaline-20min}$, and MoFe$^{Alkaline-60min}$ time points. The belt sulfurs of FeMo-cofactor S3A, S5A, and S2B are labeled.

altered conformations with respect to the P$^N$ state, with P$^{1+}$ exhibiting coordination of Fe6 to β-Ser188, and P$^{2+}$ additionally shifting the coordination of Fe5 from the thiol of α-Cys88 to its backbone amide[31]. No bridging density was observed between either the P-cluster and β-Ser188 or the α-Cys88 amide in any of the structures, thus it appears that all P-clusters are in the fully reduced P$^N$ state (Supplementary Fig. 7).

Interestingly, the highly conserved[33] β-Gln93 positioned within 6 Å of the P-cluster exhibits a conformational change reflective of the extent of disorder within the αβ-dimer. The side chain of this residue alters its rotamer in the disordered αβ dimers of the MoFe$^{Alkaline-5min}$ and MoFe$^{Alkaline-20min}$, as well as both αβ dimers of MoFe$^{Alkaline-60min}$ (Supplementary Fig. 5). Similarly, α-Trp253, which lies 12 Å from the FeMo-cofactor between the active site and the MoFe-protein surface, is

observed to flip conformations in all disordered αβ dimers, including both αβ dimers of the MoFe$^{Alkaline-60min}$ state. Computational studies have suggested that the β-Gln93 side chain may be relevant to ammonia product egress[34]. Similarly, α-Trp253 was found along a possible substrate ingress pathway as identified within xenon pressurized *A. vinelandii* MoFe-protein crystals[35]. While not as conserved as β-Gln93, the α-Trp253 residue forms a non-proline *cis*-peptide bond with the neighboring α-Ser254 residue[33,36]. The correlation between their conformation and the state of the MoFe-protein active site may support their roles in substrate and product ingress and egress[16,17].

### Uncovering a cofactor-displaced state
During processing, three dimensional variability analysis[37] (3DVA) was used to explore heterogeneity in individual cryoEM datasets (Fig. 5A–D

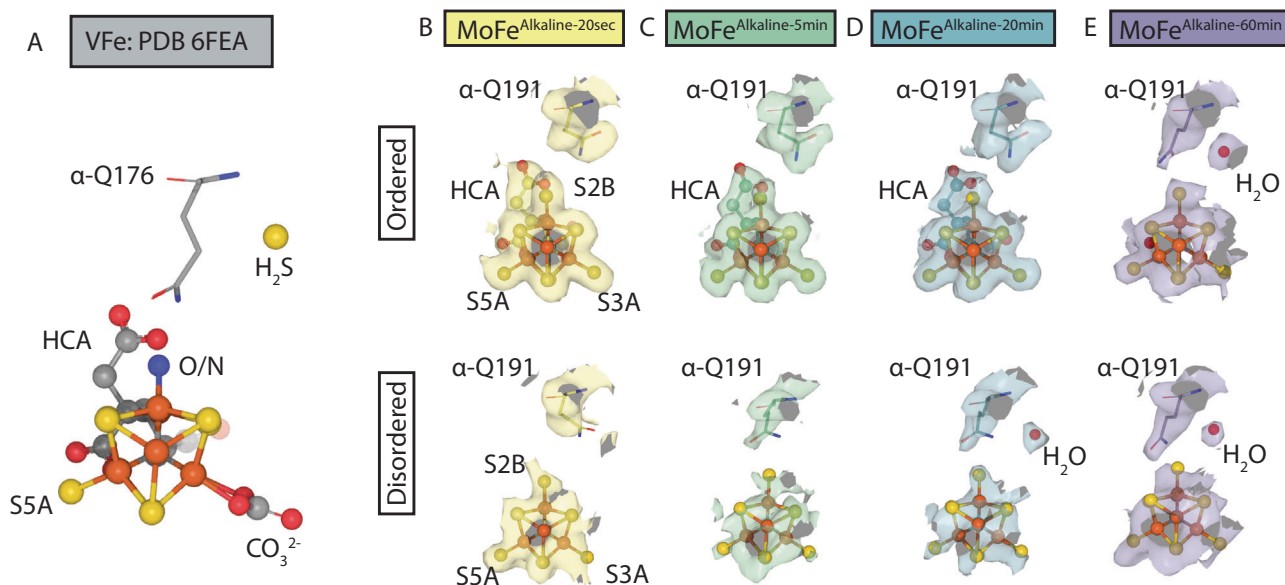

**Fig. 4 | Rearrangements in the α-Gln191 side chain. A** Previously determined crystal structure of the VFe nitrogenase for reference[15]. Residue α-Q176 in the VFe nitrogenase is the equivalent of α-Q191 in the MoFe nitrogenase. O/N oxygen or nitrogen, $H_2S$ hydrogen sulfide, $CO_3^{2-}$ carbonate. **B−E** CryoEM density surrounding the α-Q191 residue within the active sites of the alkaline turnover structures. Top panels represent ordered active sites, bottom panels represent disordered active sites.

and Supplementary Movie 1). Interestingly, subclassification with 3DVA did not reveal a symmetrically ordered resting state within any of the datasets, indicating that this state is depleted in the alkaline turnover mixtures within 20 sec. However, both MoFe^Alkaline-5min and MoFe^Alkaline-20min, but not MoFe^Alkaline-20sec, contained a class that was approximately symmetrically disordered similar to the consensus MoFe^Alkaline-60min state (Fig. 5B, C). These included MoFe^Alkaline-5min Group C and MoFe^Alkaline-20min Group A, representing 28% and 77% of the particles in each dataset, respectively. This suggests that the symmetrically disordered state seen in MoFe^Alkaline-60min accumulates in a turnover-dependent manner, perhaps reflected in the observation that the S.E.C. elution profile for each subsequent time point shifts to smaller elution volumes (Fig. 3A).

3DVA of the 5 min time point maps revealed the presence of a large density near the surface of the disordered α-subunit in a subset of particles; the same set showed a concomitant loss of density within the active site (Fig. 5B and Supplementary Movie 2). The two groups most representative of these two extremes, Group B and Group D, were subsequently locally refined about the disordered αβ dimer, resulting in two maps termed MoFe^Alkaline-5min-B and MoFe^Alkaline-5min-D at 1.97 and 2.13 Å resolution, respectively (Fig. 5E). MoFe^Alkaline-5min-D represents the most ordered subclass of the 5 min time point. Despite exhibiting disordering of α-subunit residues 1–52 and 388–417, this state still forms a cap over the active site composed of residues 353–360 and 376–387 which are missing in the more disordered states. Thus, this minimally disordered αβ dimer contains not only an ordered FeMo-cofactor in its partially disordered αβ dimer, but also an HCA molecule (Fig. 5F). In the disordered active site of the MoFe^Alkaline-5min-B state, an approximately cofactor-shaped density is observed near the surface of the protein stemming from the well-ordered loop containing α-Cys275 (Fig. 5G). The FeMo-cofactor was fit into this density and while the fit cluster shares bridging density with the α-Cys275 residue, as well as α-Ser192 and α-His195, it is not conclusive from the density how exactly the shifted cofactor might be coordinated.

### NafT binds the alkaline turnover-inactivated state

The structure of the MoFe-protein purified from HCA-deficient cells (MoFe^ΔnifV) revealed an endogenous binding interaction with the protein nitrogenase associated factor T (NafT), of unknown function[17]. Fitting the cofactor into the displaced density of the MoFe^Alkaline-5min-B state positions the cluster within 4 Å of the NafT binding interface. Given the structural similarities between MoFe^ΔnifV and the alkaline inhibited states, we sought to determine if NafT can also recognize the MoFe^Alkaline-inactivated state. Alkaline turnover reactions were performed for 4 h with and without NafT present, and subsequently subjected to concentration and S.E.C. before structural analysis (Supplementary Figs. 8 and 9). The cryoEM structure obtained from these samples revealed a 1:1 MoFe^Alkaline-inactivated-NafT complex at 2.33 Å resolution with two disordered MoFe-protein αβ-dimers. Interestingly, both active sites contain identifiable density for the FeMo-cofactors in the resting state position, despite the apparent absence of density for the HCA. This is in contrast to the previously determined MoFe^Alkaline-inactivated alone structure (PDB: 8ENL) which contained a disordered active site with no clear FeMo-cofactor density.

The interface between NafT and the MoFe-protein in the MoFe^Alkaline-inactivated-NafT structure is nearly identical to that of both the MoFe^ΔnifV-NafT (PDB: 8ENO) and MoFe^Alkaline (PDB: 8CRS) cryoEM structures, and the RMSD between these structures is 0.242 and 0.240 Å, respectively. However, within the MoFe-protein of the MoFe^Alkaline-inactivated-NafT structure, the residue α-Arg277 displays an altered conformation relative to the resting state and is positioned between the cofactor and NafT, blocking direct contact between NafT and the metallocluster (Supplementary Fig. 9). It is possible that NafT is indirectly stabilizing the FeMo-cofactor within the HCA-deficient active site through interactions at the interface between the α- and β- subunits, potentially precluding Fe-protein binding and thus preventing further turnover of a 'damaged' protein. In both αβ dimers, changes in residues α-Gln191, α-Trp253, and β-Gln93 are observed as in the disordered αβ dimers of the time resolved turnover structures, with the movement of α-Gln191 once again revealing a spherical density in the same locale as the previously identified sulfide of the VFe nitrogenase turnover structure[15] (Supplementary Fig. 9).

### Implications for the nitrogenase mechanism

The recent applications of cryoEM to the study of nitrogenase have consistently revealed large scale changes within the α-subunit of the

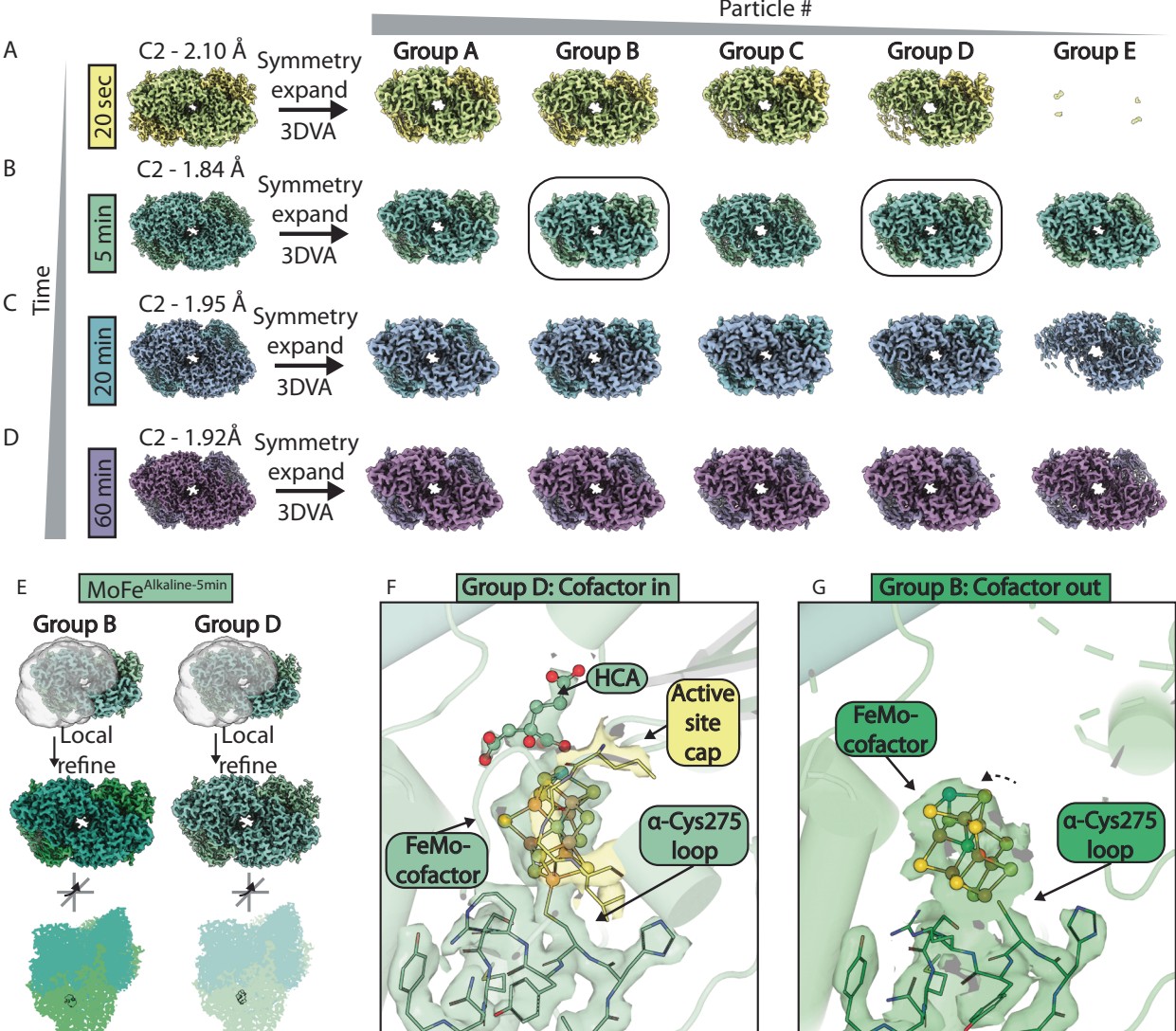

**Fig. 5 | Structure of a turnover-dependent cofactor-displaced MoFe-protein state. A–D** Left panel: CryoEM density maps of the MoFe$^{Alkaline-20sec}$ (yellow), MoFe$^{Alkaline-5min}$ (green), MoFe$^{Alkaline-20min}$ (blue), and MoFe$^{Alkaline-60min}$ (purple) time points with C2 symmetry imposed; resolution is indicated in Ångstroms. Right panel: After symmetry expansion, three dimensional variability analysis (3DVA) was carried out and clustered into 5 groups, shown from left to right in order of decreasing number of particles. The boxed groups B and D of the 5 min time point correspond to (**E–G**). **E** Groups B and D of the MoFe$^{Alkaline-5min}$ time point were subjected to local refinement with a mask covering the disordered dimer (top panel)

resulting in 1.97 Å and 2.13 Å resolution maps, respectively (middle panel). Densities for the FeMo-cofactor for each map are highlighted in the bottom panel which is related to the middle panel by ~45° rotations around both the X and Y axes. **F** CryoEM density for residues in the loop housing α-Cys275 and the FeMo-cofactor in the α-subunit of the Group D (Cofactor in) map. Residues capping the active site (355 to 358) are highlighted in yellow. **G** CryoEM density for residues in the loop housing α-Cys275 and the FeMo-cofactor in the α-subunit of the Group B (Cofactor out) map.

MoFe-protein tetramer[16,17,38]. These features have been recapitulated in our time course of structures under alkaline turnover in the presence of acetylene. By exploiting the temporal evolution of this system, we have identified a series of changes in the nitrogenase active site that suggest a highly dynamic FeMo-cofactor. Mapping these transformations along both temporal and activity axes (Fig. 6 and Supplementary Fig. 5) demonstrates that a majority of the changes we have observed occur while the MoFe-protein specific activity is still above 50% including: (1) depletion of density around the belt sulfur S2B; (2) lack of identifiable density for the HCA in the disordered active site; (3) perturbations of the inorganic framework of the FeMo-cofactor; and (4) displacement of the FeMo-cofactor from its resting state position. As acetylene binding and reduction occurs between the $E_1$ and $E_3$ states[26], we may correlate the observed changes to early steps in the catalytic cycle for substrate reduction. These observations suggest that

activation of the FeMo-cofactor for substrate binding and reduction is accompanied by changes in cofactor coordination and geometry.

This hypothesis raises a key question: how stable is the inorganic framework of the cofactor during substrate turnover? Three decades of crystallographic studies have revealed few changes within the cofactor, with the exception of the labile FeMo-cofactor belt sulfurs, notably S2B. Yet even with substitutions at these sites with CO and Se, the geometry of the cofactor remains constant[12–15,39–41]. While the changes we observe in the nitrogenase active site within this work are experimentally unprecedented, particularly alterations about the Fe7 site, several computational studies have demonstrated the possibility of expanded states, including the aforementioned McKee and Siegbahn studies[29,30]. Using QM/QM' Rao et al. suggested hydrogenation of the interstitial carbide as the first step in substrate reduction[42]. Their simulations suggested this weakens the Fe6-C bond, leading to a

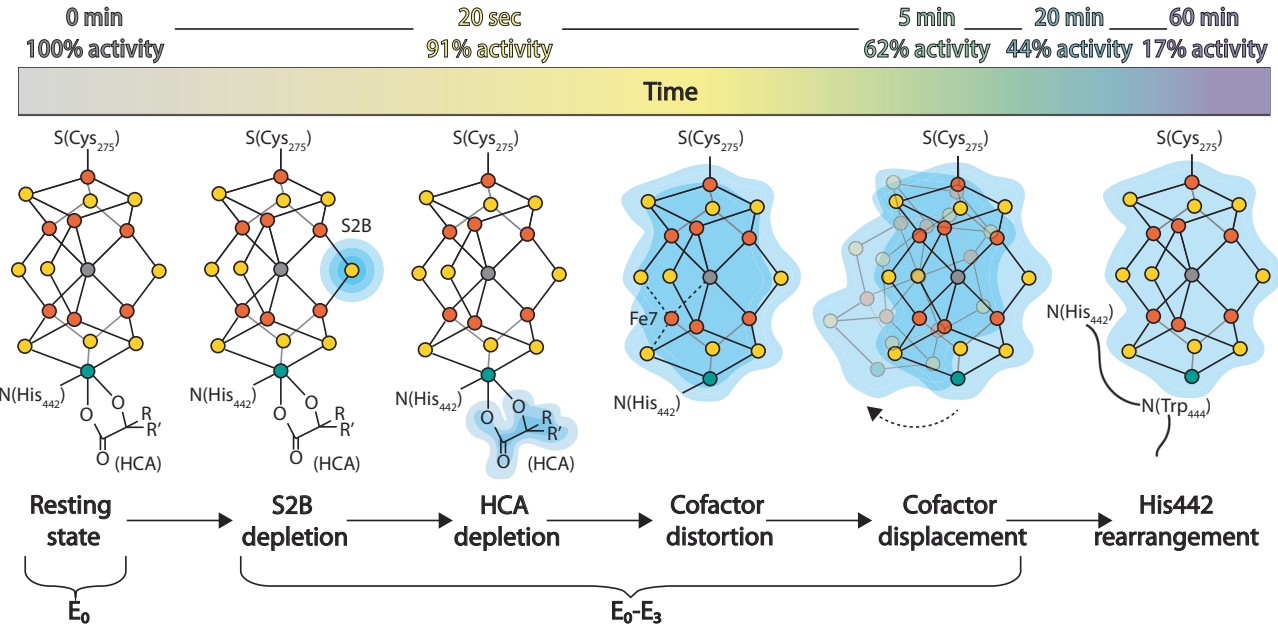

**Fig. 6 | Structural evolution of the nitrogenase active site under alkaline turnover.** Changes observed within the MoFe-protein FeMo-cofactor in cryoEM time points are diagrammed and shown from left to right as a function of time. Upper panel: Percent activity is calculated as percent remaining specific activity of acetylene reduction to ethylene at pH 7.8 from the data shown in Fig. 1C. Lower panel: Proceeding from the resting state ($E_0$), the first observed changes are depletions in the density surrounding the belt sulfur S2B (over the 20 sec to 20 min time points in the ordered active sites), which could represent positional displacement, or replacement, of this atom. Subsequently, depletions in the density at the HCA site and cofactor distortions occur at first asymmetrically between the αβ-dimers (in the 20 sec through 20 min time points) and then symmetrically (60 min time point). Within the 5 min time point, two distinct subclasses are seen either with the cofactor density displaced towards the MoFe-protein surface lacking identifiable HCA density, or in the resting state position with HCA density present. In the final time point, both cofactors occupy the resting state position, but lack identifiable HCA density and the α-His442 loop has rearranged so that the α-His442 participates in a Histidine quartet with residues α-His274, α-His362, and α-His451, while α-Trp444 instead lies adjacent to the Mo atom of the cofactor. Dashed lines between Fe7 and adjacent atoms indicate flexibility of these bonds. Color coding is as follows: Blue (cartoon representation of changing cryoEM density); yellow (sulfur atoms); orange (iron atoms); teal (molybdenum atoms); gray (carbon atoms).

structural rearrangement strongly reminiscent of the density seen within the disordered active site of the MoFe[Alkaline-20sec] structure (Fig. 3F), albeit with the outward movement of Fe6, and not Fe7 as seen here. Jiang and Ryde explored a doubly protonated interstitial carbide through computational studies, while another study from this group noted lengthening of the Fe6-C bond in more reduced states of the cofactor even in the absence of carbide protonation[43,44]. Studies of synthetic complexes further support the hypothesis that modulation of the bonding between Fe and the interstitial carbide may facilitate substrate binding and reduction[45]. Since reduction of acetylene to ethylene occurs at the $E_3$ state[26] (Supplementary Fig. 1), the alterations we observe in the cryoEM density around Fe7 support an expansion of the cofactor by this intermediate in the catalytic cycle.

In addition to distortions of the FeMo-cofactor geometry, several computational studies have pointed towards lability of the HCA ligand. Early DFT studies by Durrant suggested that the HCA coordination of the Mo atom can become monodentate upon reduction of the cluster, opening a binding site for $N_2$ at Mo[19]. Similarly, DFT studies on model complexes of metal-bound oxy-bidentate ligands proposed that protonation of this ligand could lead to chelate ring opening[46]. A more recent DFT study also suggested that a monodentate coordination of Mo by HCA was favorable in more reduced states of the cofactor[47]. It is possible that within our alkaline turnover structures, the dissociation of the HCA ligand begins with a monodentate state, which may occur on an earlier time scale than 20 sec. While our current structures do not provide clear evidence for the specific mechanism of HCA loss that is observed in the MoFe[Alkaline-inactivated] state, they emphasize the lability of the coordination of this ligand to molybdenum.

The S2B ligand has been identified as a dynamic site by multiple studies[12–15,39–41]. For the first time, we report a time-dependent change in the density corresponding to the S2B site within the well resolved, ordered active sites of the 20 sec, 5 min, and 20 min structures (Fig. 2). A much smaller decrease in cryoEM density is also noted for the S5A atom. This time-dependent decrease in S2B (and S5A) densities may be due to partial dissociation or displacement of the belt sulfurs. While a previous structural study by Sippel et al. observed a full displacement of S2B in the VFe protein, with the sulfur housed in a pocket created by the movement of the glutamine residue equivalent to α-Gln191 in the MoFe-protein, no such movement is observed within our ordered αβ dimers. Intriguingly, in the corresponding lower resolution, disordered active sites of the opposing αβ dimers in these structures, a time-dependent movement of the α-Gln191 side chain can be observed, along with the appearance of a spherical density that is near the location of the sulfur site observed in the VFe-protein (Fig. 4). Within these disordered αβ dimers, the density corresponding to the S2B atom is less interpretable, thus, the exact relationship between S2B density and the movement of the α-Gln191 side chain cannot be clearly defined within our structures. These structures do point to a time-dependent sequence of changes at these sites under acetylene turnover conditions, suggesting their relevance to the reduction mechanism of this substrate.

As acetylene reduction proceeds from the $E_3$ state of the Thorneley-Lowe kinetic model, these observations provide experimental support for the lability of S2B and distortions of the FeMo-cofactor at early steps in the substrate reduction mechanism prior to nitrogen binding. Indeed, a remarkable feature of the nitrogenase mechanism is that a relatively stable FeMo-cofactor must be activated to react with the kinetically inert $N_2$ under turnover conditions. Dissociation from α-His442 and subsequent displacement of the cofactor,

as suggested by our alkaline turnover structures, may allow for transformations at the cofactor not permitted within the constraints of the active site. For example, the observed interchange of selenium between the three belt sulfur positions[13] could be achieved through global rotation of the cofactor with respect to the protein environment, which is not possible in the resting state[17]. These rearrangements would be facilitated by a reduced coordination of the FeMo-cofactor by only two side chains, instead of the canonical one residue per metal ratio. Minimal protein coordination during turnover could thus enable cofactor rearrangement while preserving active site integrity, thereby providing a mechanism to restructure an inert cluster to catalytically reduce dinitrogen to ammonia.

## Methods

### Purification and characterization of nitrogenase proteins

The MoFe$^{As\text{-}isolated}$ and the Fe$^{As\text{-}isolated}$ proteins were purified from wild-type *A. vinelandii* Lipman (ATCC 13705, strain designation OP) under anaerobic conditions using a combination of Schlenk line techniques and anaerobic chambers with oxygen-scrubbed argon[28]. Ten L cultures of *A. vinelandii* were grown in an Eppendorf BioFlo 115 bioreactor (cat. no. BioFlo/CelliGen. 115) to an OD$_{600}$ between 1 and 2. Pelleted cells were pooled to a yield of ~50 g, and resuspended in 50 mM Tris-HCl, pH 7.8 with 150 mM NaCl and 10 mM sodium dithionite degassed buffer. Cell lysis was performed by one passage through an Emulsiflex-C5 equilibrated with anaerobic buffer at ~25,000 PSI. Cell debris was removed at 30,000 × *g* for 45 min at 4 °C. Clarified lysate was loaded onto a 70 mL QHP column equilibrated with degassed wash buffer (50 mM Tris-HCl, pH 7.8 with 150 mM NaCl and 5 mM sodium dithionite), followed by a wash step until baseline was reached. Proteins were separated using a gradient of 0–100% degassed elution buffer (50 mM Tris-HCl, pH 7.8 with 1 M NaCl and 5 mM sodium dithionite). Peaks were identified by A$_{410}$ absorbance and corresponding fractions are manually collected in sealed anaerobic Wheaton bottles. MoFe- and Fe-protein fractions were concentrated using a 50 mL Amicon filtration unit with 100 kDa and 30 kDa MWCO filters, respectively, inside an anaerobic chamber. Concentrated fractions were loaded onto a HiLoad 26/600 Superdex 200 pg column (Cytiva, cat. no. 28989336) equilibrated with degassed SEC buffer (50 mM Tris-HCl, pH 7.8 with 200 mM NaCl and 5 mM sodium dithionite). Peaks were identified by A$_{410}$ absorbance and corresponding fractions are manually collected in sealed anaerobic Wheaton bottles. Final MoFe- and Fe-protein fractions were again concentrated using a 50 mL Amicon filtration unit with 100 kDa and 30 kDa MWCO filters, respectively, inside an anaerobic chamber.

Protein concentrations were determined by absorbance at 410 nm using extinction coefficients of 76 and 9.4 mM$^{-1}$ cm$^{-1}$ for MoFe$^{As\text{-}isolated}$ and Fe$^{As\text{-}isolated}$, respectively. Concentrations in mg/mL were calculated based on molecular weights of 230 kDa and 64 kDa for MoFe$^{As\text{-}isolated}$ and Fe$^{As\text{-}isolated}$, respectively. The component ratio (CR) of Fe$^{As\text{-}isolated}$ to MoFe$^{As\text{-}isolated}$ was defined as moles of Fe$^{As\text{-}isolated}$ per mole of MoFe$^{As\text{-}isolated}$ active site via the equation CR = 1.82($C_{Fe}/C_{MoFe}$), where $C_{Fe}$ and $C_{MoFe}$ are the concentrations of the Fe-protein and MoFe-protein in the final reaction mixture, respectively, in mg/mL. Nitrogenase activity was determined by the reduction of acetylene to ethylene as measured in a 1 mL reaction volume with 9 mL headspace incubated at 30 °C for 10 min in an argon atmosphere with 1 mL acetylene gas. MoFe-protein and Fe-protein were incubated at varying CRs in reactions with 20 mM sodium dithionite, 5 mM MgCl$_2$, 5 mM ATP, 20 mM creatine phosphate, 23 U/mL creatine phosphokinase, buffered in 50 mM Tris-HCl (pH 7.8) with 200 mM NaCl. Reactions were quenched with 1 mL 3 M citric acid and ethylene was quantified by gas chromatography from 40 µL aliquots removed from the head space. The specific activity for acetylene reduction was ~3000 nmol ethylene/min/mg for MoFe$^{as\text{-}isolated}$.

### Alkaline turnover activity measurements

High pH (pH 9.5) head space reactions were performed using an ATP regeneration system of 5 mM ATP, 5 mM MgCl$_2$, and 20 mM creatine phosphate was prepared in a tribuffer system composed of 100 mM N-(2-acetamido)-2-aminoethanesulfonic acid (ACES), 50 mM tris(hydroxymethyl)aminomethane (Tris), and 50 mM ethanolamine, with pKa values of 6.67, 8.0, and 9.5 at 30 °C, respectively[24]. The pH of the ATP regeneration system was brought up to pH 9.51 with NaOH. Twenty-three U/mL phosphocreatine kinase were added, and aliquots of the solution were added to anaerobic assay vials. These solutions were made anaerobic using 12 alternating cycles of 2 min vacuum followed by 1 min Ar. After degassing, 1 mL headspace was removed from each flask and 1 mL acetylene gas was added using a Hamilton gas tight syringe model 1001. Reactions were started by the addition of 0.25 mg MoFe-protein and 0.275 mg Fe-protein (CR = 2) and were carried out in a 30 °C water bath. 50 µL aliquots of headspace was removed at the designated time points in Fig. 1C and ethylene was quantified by gas chromatography. For high pH specific activity assays measuring loss of MoFe-protein activity, initial reactions were performed identical to the head space reactions. Inactivation was monitored by the transfer of 100 µL of these initial reactions at the designated time points to a fresh reaction vial containing Fe-protein at a CR of ~60. This second reaction was allowed to proceed for 10 min at 30 °C before quenching with 1 mL 3 M citric acid. 40 µL aliquots of headspace was removed and ethylene was quantified by gas chromatography.

### Purification of recombinant NafT

The NafT sequence containing an N-terminal Strep tag (WSHPQFEK), followed by a TEV cleavage site, was cloned into a pET-21b(+) plasmid by Genscript and transformed into *Escherichia coli* BL21 cells. Four liters of *E. coli* culture containing the Strep-NafT plasmid were grown in LB broth at 37 °C with shaking at 250 rpm to 0.7–0.9 OD$_{600}$ and induced with 0.2 mM IPTG. Expression continued overnight at 17 °C and 150 RPM. Thirteen grams of pelleted cells were resuspended in 50 mL lysis buffer (100 mM Tris-HCl, pH 8, 500 mM NaCl, 20% glycerol, 1 EDTA-free Roche tablet (Sigma cat. no. 4693132001), 1 mM β-mercaptoethanol), and passed through an Emulsiflex C5 three times at 10,000 PSI. After removal of cell debris, lysate was loaded onto a 5 mL StrepTrap XT column (Cytiva cat. no. 29401322) on ice. After lysate loading, the column was washed with 100 mM Tris-HCl, pH 8, 500 mM NaCl, 20% glycerol until the 280 nm signal reached baseline. Elution of Strep-NafT was executed with a 0–100% gradient over 25 mL at 2.5 mL/min from wash buffer (100 mM Tris-HCl, pH 8, 1 M NaCl, 20% glycerol) to elution buffer (100 mM Tris-HCl, pH 8, 1 M NaCl, 20% glycerol, 50 mM biotin). Fractions containing Strep-NafT were identified by SDS-PAGE, pooled, and concentrated through a 5 kDa MWCO filter. Protein was applied to two 5 mL HiTrap desalting columns (GE healthcare cat. no. 17-1408-01) in series to remove excess salt using a desalting buffer (100 mM Tris-HCl, pH 8, 150 mM NaCl, 20% glycerol). Protein was concentrated through a 5 kDa MWCO filter. Protein concentration was determined using the TCA Lowry method[48].

### Preparation of the MoFe$^{Alkaline\text{-}inactivated}$-NafT complex

High pH (pH 9.5) turnover-inactivation reactions were performed as described above but scaled up to 20 mL volume with or without 1.8 µM Strep-NafT added. Reactions were concentrated through a 100 kDa MWCO filter using an Amicon stir cell then loaded onto a HiLoad 16/600 Superdex 200 pg (Cytiva cat. no. 28989335) S.E.C. equilibrated with 50 mM Tris-HCl, pH 7.8 with 200 mM NaCl. MoFe-protein peaks were collected at a flow rate of 1 mL/min and concentrated a second time through a 100 kDa MWCO filter. This solution was used for cryoEM grid preparation on Quantifoil R1.2/1.3 300 mesh Cu grids mixed 1:1 with 1% CHAPSO immediately before plunge freezing.

## Size exclusion chromatography of alkaline time points

For Fig. 3A, scaled alkaline acetylene reduction reaction mixtures at 20 mL final volume were prepared as described in the sections above with all concentrations being conserved. Reactions were carried out in a 30 °C water bath. At the designated time points, 1 mL of 85 mg/mL ADP in 50 mM Tris-HCl, pH 7.8 was added, and reaction flasks were cycled into an anaerobic chamber. Reactions were concentrated to ~5 mL through a 100 kDa MWCO filter using an Amicon filtration device. Concentrated reactions were loaded through a cannula onto a HiLoad 16/600 Superdex 200 pg (Cytiva cat. no. 28989335) S.E.C. equilibrated with anaerobic 50 mM Tris-HCl, pH 7.8 with 200 mM NaCl. MoFe-protein peaks were identified by $A_{410}$ and collected at a flow rate of 1 mL/min.

## Cryo-EM sample preparation and data collection

Dilutions of MoFe$^{As-isolated}$ and Fe$^{As-isolated}$ protein samples were prepared in 50 mM Tris-HCl, 200 mM NaCl, 5 mM sodium dithionite (pH 7.8) in the anaerobic chamber. Ten mL Wheaton vials containing 1 mL degassed solutions of the ATP regenerating system, 25 mM sodium dithionite (added from a concentrated stock resuspended in 0.5 M Tris, pH ~10), and 1 mL acetylene added to the headspace were cycled into the tent. Reactions were started with the addition of 0.25 mg/mL MoFe-protein and 0.275 mg/mL Fe-protein final concentration (CR = 2), and were incubated at 30 °C on a Thermo Scientific shaking dry bath (cat. no. 88880027) inside the same anaerobic chamber containing a Vitrobot Mark IV. At the defined time points, 4 µL of reaction were applied to freshly glow-discharged Quantifoil R1.2/1.3 300 mesh ultrathin carbon grids to minimize interactions with the air water interface (with the exception of the MoFe$^{Alkaline-inactivated}$-NafT complex which was prepared as described above) and blotted for 2–8 s with a blot force of 3, at ~90% humidity using the Vitrobot Mark IV[28]. Grids were stored in liquid nitrogen until data collection. Datasets were collected with a 6k x 4k Gatan K3 direct electron detector and Gatan energy filter on a 300 keV Titan Krios in superresolution mode using SerialEM 4.1.1[49] at a pixel spacing of 0.325 Å. A total dose of 60 e$^-$/Å$^2$ was utilized with a defocus range of −0.8 to −3.0 µm at the Caltech Cryo-EM Facility. The distribution of observed views for each dataset is presented in Supplementary Fig. 10.

In addition to the MoFe-protein, reaction mixtures contained the nitrogenase Fe-protein and creatine kinase for the ATP regenerating system; in total, the MoFe-protein, Fe-protein, and creatine kinase were present in reactions and on EM grids at a molar ratio of approximately 1:4:1, respectively. While distinct classes of the creatine kinase were not identified, 2D classes were obtained for the Fe-protein in all datasets and were best resolved in the 5 min time point (Supplementary Fig. 11). However, preferred orientation of the Fe-protein on the carbon substrate prevented isotropic reconstruction, likely due to adherence of the protein in a particular orientation to the carbon substrate on the EM grids. No complexes of the MoFe-protein and the Fe-protein were observed during processing, and are likely not favored at our chosen low concentration of MoFe- (~1 µM) and Fe-proteins (~4 µM). For comparison, cellular concentrations of the MoFe- and Fe-protein have been estimated to be ~26 µM and ~42 µM, respectively, in our endogenous expression system *A. vinelandii*[50].

## Cryo-EM image processing

Processing of datasets was performed in cryoSPARC v4.4 or later[51]. Movie frames were aligned and summed using patch motion correction, and the contrast transfer function (CTF) was estimated using the patch CTF estimation job in cryoSPARC. Templates for automated picking in cryoSPARC were generated by manually picking from a subset of micrographs. Picked particles were used for multiple rounds of reference-free 2D classification, ab initio model generation, and heterogeneous refinement (Supplementary Fig. 2).

Global CTF refinement was carried out, followed by non-uniform refinement[52]. Resolutions were estimated by the gold-standard Fourier shell correlation (FSC) curve with a cut-off value of 0.143. Local resolution was calculated using the local resolution estimation job in cryoSPARC. Three dimensional variability analysis[37] was carried out on 2× downsampled particles following non-uniform refinement and symmetry expansion in cryoSPARC 4.4.1 with 3 modes over 20 iterations using a filter resolution of 7 Å. Maps representing variability components were calculated using 'cluster' mode with 5 groups (clusters) and no filters. Maps corresponding to Groups B and D in the 5 min time point were re-extracted with no binning and put through local refinement with a mask surrounding the disordered αβ-dimer.

## Model building and refinement

Initial model fitting was carried out in ChimeraX 1.7[53] using PDB 3U7Q [54] for the MoFe-protein maps. The subunits are labeled such that chains A and C correspond to the ordered and disordered α-subunits, respectively. Multiple iterations were carried out of manual model building and ligand fitting in Coot 0.9.8.8[55], and refinement in Refmac5 within the CCP-EM 1.6.0 suite[56] and Phenix 1.20.1-44887-000[57]. Data collection, refinement, and validation are presented in Table S1.

## Figure preparation and presentation

Map alignments were carried out in ChimeraX 1.7 and structural figures were prepared in ChimeraX and Pymol 2.4.2[58]. Figures were compiled in Adobe Illustrator. A two phase decay model was fit to the data in Fig. 1C using GraphPad Prism version 10.0.0 for Mac, GraphPad Software, Boston, Massachusetts, USA, www.graphpad.com. GraphPad was also used for the preparation of all graphs in the article.

## Reporting summary

Further information on research design is available in the Nature Portfolio Reporting Summary linked to this article.

# Data availability

The single particle cryoEM maps and models have been deposited into the PDB and EMDB. Datasets been deposited with the following PDB and EMDB codes: 9CJE and EMD-45629, (MoFe$^{Alkaline-20sec}$), 9CJD and EMD-45628, (MoFe$^{Alkaline-5min}$), 9CJC and EMD-45627, [https://www.ebi.ac.uk/pdbe/entry/emdb/EMD-45627] (MoFe$^{Alkaline-20min}$), 9CJB and EMD-45626 (MoFe$^{Alkaline-60min}$), and 9CJF and EMD-45630 (MoFe$^{Alkaline-inactivated}$-NafT complex). CryoEM maps for 3DVA subclass maps MoFe$^{Alkaline-5min-B}$ ('cofactor out') and MoFe$^{Alkaline-5min-D}$ ('cofactor in') are included as supplementary maps with the MoFe$^{Alkaline-5min}$ deposition. PDB codes of previously published structures used in this study are 8ENL, 8ENO, 8CRS, and 3U7Q. Source Data are provided as a Source Data file. Source data are provided with this paper.

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

## Acknowledgements

This work was funded by support from the Howard Hughes Medical Institute (D.C.R.), NIH 1F32GM143836 (R.A.W.), and NIH 1K99GM152765 (R.A.W.). The generous support of the Beckman Institute for the Caltech CryoEM Resource Center was essential for the performance of this research. We thank Dr. Jens Kaiser, Dr. Songye Chen, Dr. Nathan Dalleska, and Dr. Ailiena Maggiolo for their invaluable discussions.

## Author contributions

Conceptualization: R.A.W., D.C.R. Methodology: R.A.W. Investigation: R.A.W. Visualization: R.A.W. Supervision: R.A.W., D.C.R. Writing—original draft: R.A.W. Writing—review & editing: R.A.W., D.C.R.

## Competing interests

All authors declare no competing interests.
