## [Transparent Peer Review file · Nature Communications]

Structural evolution of nitrogenase states under alkaline turnover

Corresponding Author: Dr Rebecca Warmack

Version 0:

Reviewer comments:

Reviewer #1

(Remarks to the Author)

In this study, Warmack et. al. present four different structures of Nitrogenase using a time course of alkaline reaction mixtures in the presence of acetylene. This work provides additional mechanistic information on structural changes that occur in the vicinity of the metal clusters towards nitrogen reduction. This is another elegant piece of structural work from the Rees group and builds on their ability to carry out CryoEM studies of Nitrogenase under turnover conditions.

Here, by allowing nitrogenase to turnover under alkaline conditions, a specific subset of the steps in the reduction cycle are assessed. The structural changes around the cluster are highlighted and point to the early structural perturbations in substrate turnover. We assessed all the coordinates provided by the authors and agree with the interpretations.

Since the data are compelling and the paper is well written, we only have a few minor points.

1. From a nitrogenase-expert point of view, the rationale for the work is very clear. But, for a non-expert reading this paper, an introduction figure positioning the reaction for normal N₂ reduction versus how the steps relate to acetylene under alkaline conditions might be helpful.

2. Based on the electron density, it's not very clear that there is depletion of density for S2B. Did authors perform any additional experiments for the estimation of Sulphur. Since electron density is dependent on the local resolution, how are the authors certain that the changes are due to depletion of S2B and not a reflection of poor local resolution.

3. Page 7, line 154-157, the difference in the elution volume is very subtle and is not highly significant to comment on the increase in hydrodynamic radius. Did the author perform any additional experiments to determine the hydrodynamic radius?

4. Page 4, line 76-79, the authors talk about the relatively slow nature of Nitrogenase turnover with specific numbers, but the reference is for nitrogenase from *Klebsiella pneumoniae*. Can the rate be generalized for all the nitrogenases?

5. In the CryoEM analysis, the authors started with 5 to 9 million particles for the different structures but end up with only 1.6% of the particles for 20 sec, 5.8% for 5 min, 5.6% for 20 min, 8.5% for 60 min datasets. Is there a reason for the poor percentage of particle selection?

6. In Extended Fig 3, authors have shown figure E-H but there is no information/explanation in the figure legend.

7. In Extended Fig. 3, in terms of comparing dynamics within the four structures, a common scaling would be helpful. Right now they are set differently for all the structures, a) 2-3 angstrom b) 1.5-2.5 angstrom, c) 1.75-2.75 angstrom and d) 1.5-2.5 angstrom.

8. In Extended Fig.3, A-D are colored differently which is not required for this figure as there is no specific color shown in the figures.

9. In Extended Fig.4, please add text to explain the graph and software used to prepare panel D for the general audience. Appropriate references can also be added.

10. In Fig.1, a label for the alpha-III domain is missing.
11. In Fig.2A, we can observe purple color but there is a shade of blue for 20 min figure. This Figure is not very clear, and efforts can be made to make it better.
12. In all the figures the arrow around the 90 degrees is not clear. Consider reformatting if possible.
13. Line 255, PDB ID can be added to the text for the previously determined MOFeAlkaline -inactivated structure.
14. Line 271, there is a typo 'along both along'.
15. Line 315, there is a typo 'housed a in a pocket'.
16. In the method section, Line 431, please add a reference for cryoSPARC.
17. Line 463, Fig. 1c can be corrected to bold to keep it consistent within the manuscript.
18. Figure 1. legend has a typo in Line 492, according to equation where to is missing.

Reviewer #2

(Remarks to the Author)

This manuscript by Warmack and Rees describes an informative, high-resolution series of cryoEM structures of Av MoFeP frozen during acetylene reduction at pH 7.8 at 20 sec, 5 min, 20 min, and 60 min after reaction initiation. The structures reveal very interesting, time-dependent asymmetric changes around the active sites, and they are of significance to the nitrogenase field. In the introduction, they stress that FeMoco is intrinsically dynamic, but the nature of structural changes that occur is not fully understood. They also introduce how MoFeP becomes inactivated during acetylene reduction under alkaline pH conditions, but the mechanism of deactivation is unknown. To this end, they characterized MoFeP at various timepoints corresponding to MoFeP having ~100%, 91%, 44%, and 17% activity.

The methods and analysis in this paper are rigorous. While there are some related structures that would be of interest and relevant to the work in the manuscript (such as the structure of the fully inactivated MoFeP and the timecourse structures of the FeP-MoFeP complexes from a higher FeP:MoFeP ratio), I understand that they are beyond the scope of this study. I think that the discussion is interesting, but it leaves me with some questions that I would like clarified in the manuscript. Once these questions are addressed, I recommend this manuscript for publication.

Questions about the discussion:

It is interesting that the distortions around FeMoco begin asymmetrically in just one of the $\alpha\beta$ dimers of MoFeP, and that the other dimer gradually exhibits distortions. This is especially interesting in that it corroborates the published asymmetric turnover structures. However, because the ratio of FeP:MoFeP used in these reactions was low, there were no particles containing the entire FeP:MoFeP complex. Do the authors think that the asymmetry is an intrinsic property of the acetylene reduction mechanism or rather just an asymmetry in inactivation (or both)? Because there aren't structures of the complex from this study, it is only speculation, but I think the implications of the asymmetry should be included in the discussion.

Compared to the turnover of acetylene reduction (hundreds of milliseconds), the time it takes for the distortions to occur (20 sec and longer) is quite long. Because the timescale for turnover vs the sequence of structures is so vastly different, I have some doubts that the distortions seen are directly related to structural states during turnover as opposed to structural states during inactivation that may be mutually exclusive. Maybe some of these states are comparable (and indeed many of the distortions seen are in agreement with previously published data), but I am not completely convinced that they are states E0-E3 as indicated in figure 6. Based on the turnover rate, I would expect to see a mixture of E states at 20 sec and 5 min, not see a distinct state at 20 sec and a different distinct state at 5 min. Can the authors provide clarification? Were there a mixture of states? If not, how can these states be assigned to the reaction coordinate of acetylene reduction instead of the pathway of inactivation?

Minor comments:

Figure 1 caption line 494: Should state "For D and E" rather than "For C and D".

Figure 2: It is very hard for me to distinguish between the green and blue colorings of the maps. Is there a way to make them more distinct?

Figure 5 caption: "The number of particles in the parent map are indicated below the map". I don't see the number indicated in the figure itself.

Reviewer #3

(Remarks to the Author)

This manuscript "Structural evolution of nitrogenase states under alkaline turnover" provides Cryo-EM-derived structural information of FeMo Nitrogenase during early stages of catalysis with acetylene under alkaline conditions. These conditions were employed to reduce the number of intermediate states and suppress potential side reactions resulting in Hydrogen evolution. Structures captured during these early phases are then related back to the native catalytic reactions. Important insights gained from these structures include the labile nature of a sulfide (S2B) ligand of the FeMo cofactor and a homocitrate (HCA) ligand of molybdenum. Further deconvolution analysis of Cryo-EM frames also reveals protein reorganization proximal to the active site concomitant with the lability of these ligands.

Overall, the study is well-designed and represents a significant advancement for metalloprotein enzymologists. Furthermore, the results on cofactor rearrangement would be an interesting read for the bioinorganic chemistry community. Whereas, the story is well communicated, additional commentary and slight edits would benefit the manuscript. Major and minor comments are organized below:

Major:

- The use of "alkaline pH" as a descriptor in the beginning made the text challenging to comprehend. Specifically, I would suggest rewording the sentence "Although the loss of catalytic activity at pH 9.5 follows first order kinetics with a half-life corresponding to ~10 min, the kinetics of irreversible inactivation are complex and pH dependent so that complete inactivation of the MoFe protein monitored at pH 7.8 requires 4-6 hr." I would make this sentence 2 sentences: "Although the loss of catalytic activity at pH 9.5 follows first order kinetics with a half-life corresponding to ~10 min, the kinetics of irreversible inactivation are complex and pH dependent. (In contrast) Complete inactivation of the MoFe protein monitored at pH 7.8 requires 4-6 hr."

- Similarly, I would suggest that the actual pH value be inserted into the sentence: "...directly from alkaline (pH=XX) acetylene nitrogenase turnover mixtures."

- I also had trouble with the readability of the sentence that reads "These structures are termed thestates and present views of the MoFe-protein that still retain up to 90% of their catalytic activity as measured by ethylene production at pH 7.8." Here, I would suggest splitting into two sentences. The first sentence should end at states. The second sentence should say something along the lines of "At equivalent time points and pH 7.8, the MoFe-protein that still retain up to 90% of their catalytic activity as measured by ethylene production."

- The sentence on lines 125-126 was also slightly confusing. It would be worthwhile to rephrase "a previously determined cryoEM structure of the MoFe-protein isolated from alkaline acetylene (pH =XX) in the absence of ATP, termed MoFealkaline revealed a nearly identical structure to the crystallographic resting state at physiological pH (pH = XX)."

- ATP is often known to undergo autohydrolysis to form ADP under basic conditions. Since ADP was used to quench the reduction reaction at varying timepoints when developing the structures, I'm concerned that the background production of ADP may be interfering with the catalytic process. The kinetics traces presented look solid (Fig. 1C) and the concentration of ATP (5 mM) and phosphocreatine (20 mM for ATP regeneration) could be well-saturating such that any ADP formed would not inhibit the assay or immediately be captured by phosphocreatine kinase to regenerate ATP. Additional commentary regarding this in the Methods or in an Extended Figure could help assuage my concerns.

- In Figure 3, the sulfide S5A density appears clearly present at 20 sec. but diminishes at 5 min./20 min. (Fig. 3G-H). The authors should provide some information in the methods section regarding how these maps were aligned and how appropriate based on resolution in this region it is to compare the density.

- It would be informative if the authors could also comment on the reappearance of the S5A density in the 60 min. structure. Is the density here resolved enough in these structures to comment on this sulfide's relevance for catalysis? This observation was particularly interesting given the observations made regarding the elongated distance of Fe7 to the central carbide. Could it be behaving in a similar fashion to S2B?

- A key insight from the manuscript was the lability of the HCA cofactor coordinating to Molybdenum. I assume other protein residues in the active site are aiding the positioning of this ligand via H-bonds and other electrostatic interactions. Did the structures, particularly the disordered ones where HCA was absent, reveal any reorganization of these interacting residues? If so, the authors should provide an extended data figure to show the movement of these residues across the different time point structure. This analysis would further support dynamic cofactor rearrangement mechanism the authors are proposing.

- Figure 6 and the accompanying text clearly lay out the main observations from the work. However, I'm curious about the regeneration of the cofactor after these initial catalytic steps. The authors should postulate on the regeneration of S2B or HCA cofactor to the complex. Additionally, the authors should discuss whether there is a specific order of regeneration or if the NafT protein aids in restoring the complex.

Minor:

- I was confused as to the boxed labels around the active sites of Fig. 3B-E. My best guess is that F in B = F (what is K?), H in C = G, I in D = H, J in E = I, K in E = J. These labels need to be adjusted and/or clarified in the figure legend.

- In the NaF/MoFe-inactivated structures, the authors show that Arg277 blocks access of NaF to the active site (Extended Data Fig. 8). The author should comment on whether (i) there are any new protein-protein interactions formed between the interfaces of these proteins that might be allosterically regulating MoFe and (ii) how the RMSD of MoFe residues change between NaF-bound and free structures.
- Size Exclusion Chromatography details regarding the buffer composition and flow rate should be included in the Methods.
- There is a double period in line 80
- Line 273 replace "about the belt sulfur" with around the belt sulfur
- Add the pH to line 360 "High pH (pH = XX)"
- Add the pH to line 394 "High pH (pH = XX)"
- Clarify the meaning of line 424, should it be "However, the preferred orientation of the Fe-protein prevented..."
- Add the citation to the PDB 3U7Q in line 447.

Reviewer #4

(Remarks to the Author)

Reviewer #5

(Remarks to the Author)

Version 1:

Reviewer comments:

Reviewer #1

(Remarks to the Author)

Thanks to the authors for carefully addressing our concerns. We have no further questions or concerns.

Reviewer #2

(Remarks to the Author)

I appreciate the author's response to the comments. They have clarified my concerns regarding Estates vs off-pathway inhibition conformations, and I am satisfied with their explanations. I understand that they don't wish to make further remarks in the discussion regarding some aspects of the results based on what they are unsure of (but of course I am still curious about what they think). I think that the changes that they made to the manuscript really clarify some of the key details of the work. I especially like the changes made to figure 2 based on the other reviewers' comments.

I recommend this manuscript for publication.

Reviewer #3

(Remarks to the Author)

The authors have made clear edits to the manuscript at the suggestion of all reviewers. The manuscript reads more clearly, and substantial efforts have been made to appropriately communicate their findings. We recommend the manuscript be accepted for publication.

Reviewer #4

(Remarks to the Author)

Reviewer #5

(Remarks to the Author)

REVIEWER COMMENTS

We thank the reviewers for their thoughtful and supportive comments, and in response we have modified the manuscript as detailed below, italicized and in blue.

Reviewer #1 (Remarks to the Author):

In this study, Warmack et. al. present four different structures of Nitrogenase using a time course of alkaline reaction mixtures in the presence of acetylene. This work provides additional mechanistic information on structural changes that occur in the vicinity of the metal clusters towards nitrogen reduction. This is another elegant piece of structural work from the Rees group and builds on their ability to carry out CryoEM studies of Nitrogenase under turnover conditions.

Here, by allowing nitrogenase to turnover under alkaline conditions, a specific subset of the steps in the reduction cycle are assessed. The structural changes around the cluster are highlighted and point to the early structural perturbations in substrate turnover. We assessed all the coordinates provided by the authors and agree with the interpretations.

Since the data are compelling and the paper is well written, we only have a few minor points.

1. From a nitrogenase-expert point of view, the rationale for the work is very clear. But, for a non-expert reading this paper, an introduction figure positioning the reaction for normal N₂ reduction versus how the steps relate to acetylene under alkaline conditions might be helpful.

Thank you for your suggestion, we have modified Extended Data Figure 1A and accompanying legend to clarify the rationale for the work. We have additionally modified the main text from lines 96 onward to reflect these changes.

2. Based on the electron density, it's not very clear that there is depletion of density for S2B. Did authors perform any additional experiments for the estimation of Sulphur. Since electron density is dependent on the local resolution, how are the authors certain that the changes are due to depletion of S2B and not a reflection of poor local resolution.

We thank the reviewer for this comment. As there is no equivalent experiment in electron microscopy to the anomalous scattering in X-ray crystallography, we were not able to perform additional experiments to confirm the loss of sulfur. However, we have modified Figure 2 to include the cryoEM density, as well as the volume measurements, for the additional belt sulfurs S3A and S5A in the ordered active sites. As the local resolution of the cofactor is quite good (est. ~1.6 Å by local resolution maps, Extended Data Figure 3), we can see from these additional measurements that while the volumes

of S3A and S5A remain relatively constant, the volume of the density surrounding S2B decreases steeply by comparison.

3. Page 7, line 154-157, the difference in the elution volume is very subtle and is not highly significant to comment on the increase in hydrodynamic radius. Did the author perform any additional experiments to determine the hydrodynamic radius?

Thank you for this comment, we have not performed additional experiments to determine the hydrodynamic radius. Although the shift is small, we have found that the change in elution time is highly reproducible since the first observation in Yang et al., 2014. We have hypothesized that the change in elution time reflects a change in the hydrodynamic radius, but we have not established this experimentally. In the manuscript we have tried to capture this uncertainty through more qualified language in line 193.

4. Page 4, line 76-79, the authors talk about the relatively slow nature of Nitrogenase turnover with specific numbers, but the reference is for nitrogenase from *Klebsiella pneumoniae*. Can the rate be generalized for all the nitrogenases?

We have included the reference Eady 1972 and included additional text in Lines 78-79 to clarify that the Kp and Av enzymes have similar specific activities.

5. In the CryoEM analysis, the authors started with 5 to 9 million particles for the different structures but end up with only 1.6% of the particles for 20 sec, 5.8% for 5 min, 5.6% for 20 min, 8.5% for 60 min datasets. Is there a reason for the poor percentage of particle selection?

We apologize for the confusion. In our experience, the template picking picks large numbers of spurious peaks, resulting in an inflated 'initial' particle number. We believe a better representation of the initial nitrogenase particle population is the initially selected 2D classes, thus we have included these numbers in Extended Data Figure 2. The average percentage of MoFe-protein particles used based on this initial number is 29%. This also tracks with the mixture of proteins that are present on the grid, as our ratio of MoFe-protein to Fe-protein to creatine kinase is approximately 1:4:1.

6. In Extended Fig 3, authors have shown figure E-H but there is no information/explanation in the figure legend.

Thank you for pointing out this issue, we have now edited the figure legend to clarify the information for panels E-H.

7. In Extended Fig. 3, in terms of comparing dynamics within the four structures, a common scaling would be helpful. Right now they are set differently for all the structures, a) 2-3 angstrom b) 1.5-2.5 angstrom, c) 1.75-2.75 angstrom and d) 1.5-2.5 angstrom.

We did try a common scale for the four structures, however, we found that this resulted in certain maps being displayed mostly in one color. The 20 sec map was particularly

challenging. We believe the differing scales allow for important internal comparisons of the local resolutions. As an example, the comparison of the local resolution around the FeMo-cofactor within the disordered active site of the 20 sec time point vs. that in the ordered active site relies on a scale that highlights the differing local resolution between the two. This information was lost when we tried to use a common scale across all four structures.

8. In Extended Fig.3, A-D are colored differently which is not required for this figure as there is no specific color shown in the figures.

We thank the author for this comment. While the color scheme is not employed in this figure, we left these labels included in the figure to reinforce the color scheme used throughout the manuscript.

9. In Extended Fig.4, please add text to explain the graph and software used to prepare panel D for the general audience. Appropriate references can also be added.

We have clarified this panel and expanded the information in our "Figure preparation and presentation" section of the methods to indicate that GraphPad was used to prepared this figure.

10. In Fig.1, a label for the alpha-III domain is missing.

We have labeled the alpha and beta subunits within figure 1, but we have left out a label for the alpha III domain to avoid an overwhelming number of colors and labels in figure 1. We have outlined the alpha III domain in extended data figure 4, which is referenced in the main text section discussing figure 1.

11. In Fig.2A, we can observe purple color but there is a shade of blue for 20 min figure. This Figure is not very clear, and efforts can be made to make it better.

We have edited this figure so that each panel is separate.

12. In all the figures the arrow around the 90 degrees is not clear. Consider reformatting if possible.

We have reformatted the arrows in the main figures.

13. Line 255, PDB ID can be added to the text for the previously determined MOFeAlkaline - inactivated structure.

We have added this PDB code to line 305.

14. Line 271, there is a typo 'along both along'.

We have addressed this typo.

15. Line 315, there is a typo 'housed a in a pocket'.

We have addressed this typo.

16. In the method section, Line 431, please add a reference for cryoSPARC.

We have added this citation.

17. Line 463, Fig. 1c can be corrected to bold to keep it consistent within the manuscript.

We have bolded this text.

18. Figure 1. legend has a typo in Line 492, according to equation where to is missing.

We have addressed this typo.

Reviewer #2 (Remarks to the Author):

This manuscript by Warmack and Rees describes an informative, high-resolution series of cryoEM structures of Av MoFeP frozen during acetylene reduction at pH 7.8 at 20 sec, 5 min, 20 min, and 60 min after reaction initiation. The structures reveal very interesting, time-dependent asymmetric changes around the active sites, and they are of significance to the nitrogenase field. In the introduction, they stress that FeMoco is intrinsically dynamic, but the nature of structural changes that occur is not fully understood. They also introduce how MoFeP becomes inactivated during acetylene reduction under alkaline pH conditions, but the mechanism of deactivation is unknown. To this end, they characterized MoFeP at various timepoints corresponding to MoFeP having ~100%, 91%, 44%, and 17% activity.

The methods and analysis in this paper are rigorous. While there are some related structures that would be of interest and relevant to the work in the manuscript (such as the structure of the fully inactivated MoFeP and the timecourse structures of the FeP-MoFeP complexes from a higher FeP:MoFeP ratio), I understand that they are beyond the scope of this study. I think that the discussion is interesting, but it leaves me with some questions that I would like clarified in the manuscript. Once these questions are addressed, I recommend this manuscript for publication.

Questions about the discussion:

It is interesting that the distortions around FeMoco begin asymmetrically in just one of the $\alpha\beta$ dimers of MoFeP, and that the other dimer gradually exhibits distortions. This is especially interesting in that it corroborates the published asymmetric turnover structures. However, because the ratio of FeP:MoFeP used in these reactions was low, there were no particles containing the entire FeP:MoFeP complex. Do the authors think that the asymmetry is an intrinsic property of the acetylene reduction mechanism or rather just an asymmetry in inactivation (or both)? Because there aren't structures of the complex from this study, it is only speculation, but I think the implications of the asymmetry should be included in the discussion.

We have emphasized the asymmetry of these structures within the first results section, and noted the similarity to the asymmetry seen within the structures of the N2 turnover studies of Rutledge et al. We do believe that asymmetry and disorder within the alpha subunit are intrinsic properties of nitrogenase turnover, which have been discussed in the previous Rutledge et al and Warmack et al 2023 publications. The implications of these properties for the mechanism are still unclear, at least in our view, and so we have not further highlighted these attributes in the discussion.

Compared to the turnover of acetylene reduction (hundreds of milliseconds), the time it takes for the distortions to occur (20 sec and longer) is quite long. Because the timescale for turnover vs the sequence of structures is so vastly different, I have some doubts that the distortions seen are directly related to structural states during turnover as opposed to structural states during inactivation that may be mutually exclusive. Maybe some of these states are comparable (and indeed many of the distortions seen are in agreement with previously published data), but I am not completely convinced that they are states E0-E3 as indicated in figure 6. Based on the turnover rate, I would expect to see a mixture of E states at 20 sec and 5 min, not see a distinct state at 20 sec and a different distinct state at 5 min. Can the authors provide clarification? Were there a mixture of states? If not, how can these states be assigned to the reaction coordinate of acetylene reduction instead of the pathway of inactivation?

We agree that there are a mixture of states present, and demonstrate support for this hypothesis through the 3DVA shown in Figure 5, which highlights that nearly all 4 structures are a mixture of different states including asymmetrically and symmetrically disordered states. Based on the activity measurements at pH 9.5 in Fig. 1C, there is roughly 1 nmol ethylene produced by 1 nmol MoFe-protein at 20 sec. Therefore, the reaction is still in the initial phase and it seems unlikely that steady state has been achieved at this first time point. We have added text to the first results section, starting at line 120, to clarify these points, as well as the legend of Figure 1 (lines 574-575).

Additionally, ~91 and 62% of activity are still retained at the 20 sec and 5 min time points, respectively, as measured at pH 7.8. Thus despite the slow inhibition of MoFe-protein at alkaline pH, these first time points should still be dominated by active material. We emphasize that the mechanism of inactivation is turnover-dependent, so the inactivated states would stem from those seen in the earlier time points, and we would not expect the inactivated material to be dominant until the later time points where activity has decreased to less than 50%.

Minor

comments:

Figure 1 caption line 494: Should state "For D and E" rather than "For C and D".

We have addressed this typo.

Figure 2: It is very hard for me to distinguish between the green and blue colorings of the maps. Is there a way to make them more distinct?

We have edited figure 2 so that there are separate panels displaying density, making the colors easier to distinguish. We specifically used the Viridis color palette for its accessibility to color blind readers.

Figure 5 caption: "The number of particles in the parent map are indicated below the map". I don't see the number indicated in the figure itself.

We apologize for the confusion, we have removed this line from the manuscript.

Reviewer #3 (Remarks to the Author):

This manuscript "Structural evolution of nitrogenase states under alkaline turnover" provides Cryo-EM-derived structural information of FeMo Nitrogenase during early stages of catalysis with acetylene under alkaline conditions. These conditions were employed to reduce the number of intermediate states and suppress potential side reactions resulting in Hydrogen evolution. Structures captured during these early phases are then related back to the native catalytic reactions. Important insights gained from these structures include the labile nature of a sulfide (S₂B) ligand of the FeMo cofactor and a homocitrate (HCA) ligand of molybdenum. Further deconvolution analysis of Cryo-EM frames also reveals protein reorganization proximal to the active site concomitant with the lability of these ligands. Overall, the study is well-designed and represents a significant advancement for metalloprotein enzymologists. Furthermore, the results on cofactor rearrangement would be an interesting read for the bioinorganic chemistry community. Whereas, the story is well communicated, additional commentary and slight edits would benefit the manuscript. Major and minor comments are organized below:

Major:

- The use of "alkaline pH" as a descriptor in the beginning made the text challenging to comprehend. Specifically, I would suggest rewording the sentence "Although the loss of catalytic activity at pH 9.5 follows first order kinetics with a half-life corresponding to ~10 min, the kinetics of irreversible inactivation are complex and pH dependent so that complete inactivation of the MoFe protein monitored at pH 7.8 requires 4-6 hr." I would make this sentence 2 sentences: Although the loss of catalytic activity at pH 9.5 follows first order kinetics with a half-life corresponding to ~10 min, the kinetics of irreversible inactivation are complex and pH dependent. (In contrast) Complete inactivation of the MoFe protein monitored at pH 7.8 requires 4-6 hr.

Thank you for this suggestion, we have incorporated this wording in the text.

- Similarly, I would suggest that the actual pH value be inserted into the sentence: "...directly from alkaline (pH=XX) acetylene nitrogenase turnover mixtures."

We have incorporated this suggestion.

- I also had trouble with the readability of the sentence that reads "These structures are termed thestates and present views of the MoFe-protein that still retain up to 90% of their catalytic activity as measured by ethylene production at pH 7.8." Here, I would suggest splitting into two sentences. The first sentence should end at states. The second sentence should say something along the lines of "At equivalent time points and pH 7.8, the MoFe-protein that still retain up to 90% of their catalytic activity as measured by ethylene production."

We have split this sentence into two parts.

- The sentence on lines 125-126 was also slightly confusing. It would be worthwhile to rephrase "a previously determined cryoEM structure of the MoFe-protein isolated from alkaline acetylene (pH =XX) in the absence of ATP, termed MoFealkaline revealed a nearly identical structure to the crystallographic resting state at physiological pH (pH = XX).

We have reworded this sentence.

- ATP is often known to undergo autohydrolysis to form ADP under basic conditions. Since ADP was used to quench the reduction reaction at varying timepoints when developing the structures, I'm concerned that the background production of ADP may be interfering with the catalytic process. The kinetics traces presented look solid (Fig. 1C) and the concentration of ATP (5 mM) and phosphocreatine (20 mM for ATP regeneration) could be well-saturating such that any ADP formed would not inhibit the assay or immediately be captured by phosphocreatine kinase to regenerate ATP. Additional commentary regarding this in the Methods or in an Extended Figure could help assuage my concerns.

We thank the reviewer for highlighting this confusing point – the structures were not prepared in the presence of ADP, but rather ADP was used as a quenching agent in the case of the size exclusion chromatography to isolate these states throughout the long time course of that experiment. The structures were determined straight from reaction mixtures containing ATP and the ATP regeneration system with no ADP present. To clarify this point we have reworded the sentence in lines 191-193.

- In Figure 3, the sulfide S5A density appears clearly present at 20 sec. but diminishes at 5 min./20 min. (Fig. 3G-H). The authors should provide some information in the methods section regarding how these maps were aligned and how appropriate based on resolution in this region it is to compare the density.

We agree that low local resolution within the disordered active sites prevents definitive comparisons from being made at these sites, and we have not made conclusions within the text regarding belt sulfur occupancy in the disordered active sites. We have added information on line 542 of the methods regarding map alignments.

- It would be informative if the authors could also comment on the reappearance of the S5A density in the 60 min. structure. Is the density here resolved enough in these structures to comment on this sulfide's relevance for catalysis? This observation was particularly interesting given the observations made regarding the elongated distance of Fe7 to the central carbide. Could it be behaving in a similar fashion to S2B?

We have edited the main text from lines 222-224 to comment on the S5A density, while leaving in the qualification that "the local resolution limits further interpretation." We also clarify that the reappearance of the density within the 60 min structure is likely due to the map being a composite of a more homogeneous mixture of particles, which is supported by the 3DVA results presented in figure 4 that demonstrate nearly all subclasses of the 60 min time point as being symmetric, while the 5 min and 20 min time points appear to be more heterogeneous.

- A key insight from the manuscript was the lability of the HCA cofactor coordinating to Molybdenum. I assume other protein residues in the active site are aiding the positioning of this ligand via H-bonds and other electrostatic interactions. Did the structures, particularly the disordered ones where HCA was absent, reveal any reorganization of these interacting residues? If so, the authors should provide an extended data figure to show the movement of these residues across the different time point structure. This analysis would further support dynamic cofactor rearrangement mechanism the authors are proposing.

Thank you for your comment. We have shown the movements of all residues with sufficient density that have notable changes in Extended Data Figure 5, including residues near the HCA like His442 and Gln191. Importantly, we show in the main text Figure 5 that between the two subclasses of the 5 min time point in which we observe putative cofactor in and out states, there appear to be a crucial set of residues missing density within the cofactor out state. We term these residues the 'active site cap', which may keep the cofactor and the HCA within the active site. This is discussed on lines 283-285.

- Figure 6 and the accompanying text clearly lay out the main observations from the work. However, I'm curious about the regeneration of the cofactor after these initial catalytic steps. The authors should postulate on the regeneration of S2B or HCA cofactor to the complex. Additionally, the authors should discuss whether there is a specific order of regeneration or if the NafT protein aids in restoring the complex.

From our studies, it is not clear whether or not the belt sulfur fully dissociates from the cofactor under turnover or whether it may remain bound to one iron. We have discussed the possible role of NafT's involvement in stabilizing the alkaline inactivated state of nitrogenase from lines 313-316 in the main text.

Minor:

- I was confused as to the boxed labels around the active sites of Fig. 3B-E. My best guess is

that F in B = F (what is K?), H in C = G, I in D = H, J in E = I, K in E = J. These labels need to be adjusted and/or clarified in the figure legend.

We apologize for this error, the lettering of these panels has been corrected.

- In the NafT/MoFe-inactivated structures, the authors show that Arg277 blocks access of NafT to the active site (Extended Data Fig. 8). The author should comment on whether (i) there are any new protein-protein interactions formed between the interfaces of these proteins that might be allosterically regulating MoFe and (ii) how the RMSD of MoFe residues change between NafT-bound and free structures.

We have modified lines 307 to 310 to include discussion of the interfaces between NafT and the MoFe-protein, as well as the RMSD values.

- Size Exclusion Chromatography details regarding the buffer composition and flow rate should be included in the Methods.

We have added a separate SEC section within the methods.

- There is a double period in line 80

This typo has been addressed.

- Line 273 replace "about the belt sulfur" with around the belt sulfur

This has been changed.

- Add the pH to line 360 "High pH (pH = XX)"

Changed.

- Add the pH to line 394 "High pH (pH = XX)"

Changed.

- Clarify the meaning of line 424, should it be "However, the preferred orientation of the Fe-protein prevented..."

We have modified this sentence.

- Add the citation to the PDB 3U7Q in line 447.

This citation has been added.